# Interface Effects on the Electronic and Optical Properties of Graphitic Carbon Nitride (g-C_3_N_4_)/SnS_2_: First-Principles Studies

**DOI:** 10.3390/ma18040892

**Published:** 2025-02-18

**Authors:** Li-Hua Qu, Yu Wang, Si-Wen Xia, Ran Nie, Le Yin, Chong-Gui Zhong, Sheng-Li Zhang, Jian-Min Zhang, You Xie

**Affiliations:** 1School of Physical Science and Technology, Nantong University, Nantong 226019, China; yw_1217@163.com (Y.W.); shawell12138@163.com (S.-W.X.); nieran0316@163.com (R.N.); 2002110078@stmail.ntu.edu.cn (L.Y.); chgzhong@ntu.edu.cn (C.-G.Z.); 2Nanjing University of Science and Technology, Nanjing 210094, China; zhangslvip@njust.edu.cn; 3College of Physics and Information Technology, Shaanxi Normal University, Xi’an 710062, China; jmzhang@snnu.edu.cn; 4College of Science, Xi’an University of Science and Technology, Xi’an 710054, China; xieyou@xust.edu.cn

**Keywords:** interface effects, electronic properties, optical properties, G-C_3_N_4_/SnS_2_, first principles

## Abstract

Heterojunctions have received much interest as a way to improve semiconductors’ electrical and optical properties. The impact of the interface on the electrical and optical properties of g-C_3_N_4_/SnS_2_ was explored using first-principles calculations in this study. The results show that, at the hetero-interface, a conventional type-II band forms, resulting in a lower band gap than that in the g-C_3_N_4_ and SnS_2_ monolayers. When there is no high barrier height, the averaged microscopic and averaged macroscopic potentials can be used to accomplish efficient carrier transformation. Furthermore, the polarization direction affects the absorption spectra. All of these discoveries have significant implications for the development of g-C_3_N_4_-based optoelectronics.

## 1. Introduction

Semiconductor photocatalysis is a green technology that provides significant social and economic benefits and has a wide range of applications in antisepsis, water quality improvement, and environmental optimization [1,2,3]. In recent years, it has become increasingly vital to investigate effective photocatalysts. Since Wang demonstrated that g-C_3_N_4_ can be photocatalytically oxidized and create H_2_ under visible light [4], g-C_3_N_4_ photocatalyst research has received much attention. Because of its high stability and sufficient band gap [5,6], g-C_3_N_4_ is a better contender among the many photocatalysts available. However, due to the high photogenerated electron–hole recombination rate and poor light absorption caused by the low specific surface area, pure g-C_3_N_4_ has a low photocatalytic efficiency. As a result, several researchers, including those concerned with g-C_3_N_4_ functionalization and chemical doping, have concentrated their efforts on improving electron–hole separation and visible optical absorption, which are crucial for improving catalytic efficiency [5,7]. The photocatalytic effectiveness of heterojunctions is shown to be higher than that of previous modification plans. As a result, scientists have worked on a variety of g-C_3_N_4_/semiconductor heterojunctions, including g-C_3_N_4_/SnO_2_ [8], g-C_3_N_4_/TiO_2_ [9], g-C_3_N_4_/Zn_2_GeO_4_ [10], and NiO/g-C_3_N_4_ [11].

The n-type semiconductor SnS_2_ has sparked curiosity because of its low price, nontoxicity, and acceptable band gap (approximately 2.4 eV), all of which make it ideal for photocatalysis. The band edge potentials of SnS_2_ and g-C_3_N_4_ have been found to be extremely comparable [12]. As a result, modeling the heterojunction can be very useful in photocatalytic activity enhancement. G-C_3_N_4_/SnS_2_ has been experimentally synthesized, and its remarkable photocatalytic activity in visible light has been demonstrated [13].

Inspired by this experimental idea, we attempt to study the hybridization between the SnS_2_ sheet and g-C_3_N_4_. To design and increase the photocatalytic effectiveness of the g-C_3_N_4_/SnS_2_ heterostructure, the interfacial effect on the structural, electronic, and optical properties must be studied. Herein, based on density functional theory (DFT) calculations, we try to investigate the interfacial interaction between monolayer g-C_3_N_4_ and SnS_2_ sheet and band structures of g-C_3_N_4_ and SnS_2_. The computation results show a type-II band alignment, which is desirable for visible light water splitting. Moreover, due to the charge transfer between g-C_3_N_4_ and SnS_2_ sheet, it can further enhance the photogenerated electron–hole separation. These results suggest that SnS_2_ sheet would be a nice choice to improve the photocatalytic performance of g-C_3_N_4_ photocatalyst.

## 2. Computational Methods

The Vienna ab initio Simulation Package (VASP 5.4, Austrian) was used to execute the computations, which were based on the projector augmented wave (PAW) approach and included dispersion correction (DFT-D2) [14,15,16,17,18,19,20]. The exchange and correlation potentials are described using the Perdew−Burke−Ernzerhof (PBE) method of generalized gradient approximation (GGA) [16]. When the energy and force on each ion are decreased below 10^−6^ eV and 0.02 eV/Å, structural optimization occurs. The cutoff energy of the plane-wave base group is set to 520 eV, and a Monkhorst–Pack k-grid of 7 × 7 × 1 is utilized [21].

The charge density difference is described as follows to understand how charge transfer occurs [22]:(1)$$\rho =\rho_{g-C_{3}N_{4}/SnS_{2}}-\rho_{g-C_{3}N_{4}}-\rho_{SnS_{2}}$$
where $\rho_{g-C_{3}N_{4}/SnS_{2}}$, $\rho_{g-C_{3}N_{4}}$, and $\rho_{SnS_{2}}$ are the charge densities of g-C_3_N_4_/SnS_2_, g-C_3_N_4_, and SnS_2_ monolayers in one system, respectively. Positive and negative numbers represent charge accumulation and depletion, respectively.

The interfacial valence electron change is explored to see if the interfacial contact is mostly physisorption. Then, the integrated charge density difference throughout the x–y plane is then integrated along the z direction, which is perpendicular to the interface, and the plane-averaged charge difference Δq [23] is defined as follows:(2)$$\Delta q=\int_{-\infty }^{+\infty }\int_{-\infty }^{+\infty }\left(\rho_{SnS_{2}/g-C_{3}N_{4}}-\rho_{SnS_{2}}-\rho_{g-C_{3}N_{4}}\right)dxdy$$

Furthermore, by integrating along the *z*-axis, the charge displacement curve (CDC) is defined by integrating ΔQ [23]:(3)$$\Delta Q=\int_{-\infty }^{z}\Delta qdz$$

## 3. Results and Discussion

### 3.1. Geometric Structures

In our work, we perform benchmark calculations for the isolated single-layer g-C_3_N_4_ and SnS_2_ sheets to examine the reliability of the computational parameters. After full relaxation, the lattice parameters are a = b = 7.15 Å for g-C_3_N_4_, and a = b = 3.70 Å for SnS_2_. To build a g-C_3_N_4_/SnS_2_ heterojunction, we adopt 1 × 1 and 2 × 2 supercells for g-C_3_N_4_ (6 C and 15 N atoms) and SnS_2_ (including 4 Sn and 8 S atoms), respectively, as shown in Figure 1. For g-C_3_N_4_ and SnS_2_, the mismatch is 3% for the x and y directions, and a vacuum spacing of 15 is used to reduce artificial contact. For this heterogeneous structure, we perform full structure relaxation, and the lattice parameter is a = b = 7.18 Å; meanwhile, the minimum spacing between two monolayers is 3.06 Å. The bond lengths of C1-N1, C2-N2, C3-N3, SnS1, and Sn-S2 are 1.490, 1.353, 1.348, 2.568, and 2.567 Å, respectively.

These findings are comparable with those obtained with g-C_3_N_4_ and SnS_2_ monolayers, showing a modest contact between the two. The interfacial formation energy ($E_{F}$) [24] is computed using the following equation to determine stability:(4)$$E_{F}=E_{g-C_{3}N_{4}/SnS_{2}}-E_{g-C_{3}N_{4}}-E_{SnS_{2}}$$
where $E_{g-C_{3}N_{4}/SnS_{2}}$, $E_{g-C_{3}N_{4}}$, and $E_{SnS_{2}}$ are the total energies of the heterojunction and monolayers. The heterojunction’s interfacial formation energy is predicted to be −0.027 eV, demonstrating improved thermodynamic stability. Furthermore, the phonon spectrum of the heterostructure is shown in Appendix A to elucidate stability.

To investigate the optical properties of g-C_3_N_4_/SnS_2_, the optical absorption spectra are calculated by converting the complex dielectric function to the absorption coefficient α by the following equation [25], where $\varepsilon_{1}$ and $\varepsilon_{2}$ are the real and imaginary absorption factors, respectively.(5)$$\alpha =\frac{4\pi E}{hc}{\left(\frac{\sqrt{\varepsilon_{1}^{2}+\varepsilon_{2}^{2}}-\varepsilon_{1}}{2}\right)}^{\frac{1}{2}}$$

### 3.2. Fat Band and Density of States (DOS)

As illustrated in Figure 2, the fat bands and DOS are estimated to indicate high photocatalytic activity. The conduction band minimum (CBM) and valence band maximum (VBM) of g-C_3_N_4_/SnS_2_ are located at the Γ and M points, respectively, in Figure 2a, providing a narrower gap than that of the monolayers (2.197 and 2.864 eV in Figure 2b,c).

The main features of the DOS of Sn, S, N, and C atoms in the g-C_3_N_4_/SnS_2_ heterostructure are similar to the results of the isolated g-C_3_N_4_ and SnS_2_. This is because of the weak interaction and relatively large separation between the g-C_3_N_4_ and SnS_2_. As can be seen in Figure 2a, SnS_2_ (g-C_3_N_4_) contributes to the CBM (VBM) of the heterojunction system. The CBM (VBM) largely disperses over the p states of N and C atoms (p and d states of S and Sn atoms, respectively) in Figure 2b,c. Moreover, the VBM of g-C_3_N_4_ is 0.22 higher than that of SnS_2_, suggesting the interfacial type-II band alignment of the g-C_3_N_4_/SnS_2_ heterostructure, in which the valence band offset (VBO) between g-C_3_N_4_ and SnS_2_ is about 0.22 eV, and the conduction band offset (CBO) is about 1.01 eV. This will be addressed further in the subsequent charge transport investigation. Therefore, it can be concluded that photocatalysis will transport electrons from SnS_2_ to g-C_3_N_4_, resulting in the redox process.

### 3.3. Charge Transport

The charge redistribution is mainly distributed in the interfacial regions (positive and negative values are expressed as charge accumulation (yellow isosurface) and depletion (blue isosurface) in the adhesion process, respectively), as shown in Figure 3a,b, which can be explained by (i) minor local lattice distortion due to a small lattice mismatch; (ii) the high-strength double-layer honeycomb structure; and (iii) the elimination of lattice faults when the interlayer gap between SnS_2_ and g-C_3_N_4_ is large enough. The averaged microscopic and averaged macroscopic potentials are then determined, as shown in Figure 3c. It shows that there is no high barrier height to influence the carrier transformation. Despite the electron transfer (positive value of the charge displacement curve (CDC, ΔQ) in the interfacial region) during the creation of g-C_3_N_4_/SnS_2_, the x–y plane average charge density difference Δq suggests that no charge accumulates near the g-C_3_N_4_ layer side, as shown by the difference in the average charge density on the x–y plane q. Therefore, when the lattice mismatch is modest, the structure is strong, and the interfacial defects are minor. Smaller local lattice distortion, lesser interfacial defect, and thin interfacial regions can better hold the physicochemical properties of the component, and the physicochemical parameters of the heterojunction can be easily controlled, limiting the impacts of interfacial trapping states and transition barriers on carrier transport.

The localized PDOS in the normal direction of the g-C_3_N_4_/SnS_2_ interfaces (x–y plane) is depicted in Figure 3d to study charge transport channels at the interfaces. At the heterojunction interfaces, a staggered-gap (type II) offset junction is plainly visible, implying that the charge transfer is fundamentally justified. There is no discernible difference between the g-C_3_N_4_ and SnS_2_ monolayers when the band gap and interface are separated by a greater distance. Close to the interface, however, it drops. When a photon of light is absorbed by g-C_3_N_4_, it energizes electrons and holes, which are then negatively transported down the CBM and VBM. That is to say, the spontaneous interfacial charge transfer from the g-C_3_N_4_ to SnS_2_ can be imply rationalized in terms of the large difference between two layers, which results in a built-in electric field at the interface, promoting photogenerated electrons from the CB of SnS_2_ to the CB of g-C_3_N_4_, and improving the transformation of photogenerated holes of the VB from g-C_3_N_4_ to SnS_2_. Therefore, the photogenerated electron–hole separation can be enhanced effectively by this built-in interface electric field. The yellow shadow in Figure 3e represents the interface states that emerge at the margin of the g-C_3_N_4_ conduction band. Due to band edge rearrangement and redistribution, the interface states minimize the heterojunction system’s band gap, promoting efficient charge extraction and providing a viable way to design efficient g-C_3_N_4_-/SnS_2_-based optoelectronic devices.

### 3.4. Optical Properties

The photocatalytic efficiency of the g-C_3_N_4_ monolayer and g-C_3_N_4_/SnS_2_ can be calculated using the absorption spectra shown in Figure 4. The peak energy of g-C_3_N_4_ is around 3.3 eV, as shown in the image. In contrast, the visible absorption spectra of the heterojunction are successfully improved because the absorption edge changes from UV to the near-infrared region, demonstrating polarization-dependent behavior. The optical absorption coefficients in the α direction parallel to the x–y plane are greater than those in the perpendicular direction, as shown in Figure 4.

## 4. Conclusions

In summary, first-principles DFT simulations were used in this study to investigate the impact of the interface on the electrical and optical properties of the organic–inorganic g-C_3_N_4_/SnS_2_. Because the heterojunction system has a band alignment of type II, it has a smaller band gap than that of g-C_3_N_4_ and SnS_2_ monolayers. The built-in interface electric field within the interface region is desirable for photogenerated carrier separation. The averaged microscopic and averaged macroscopic potentials reveal that carrier transformation is achievable even when the barrier height is low. The electric field and the nice band edge alignment suggest that the g-C_3_N_4_/SnS_2_ heterostructure will have good application prospects in the field of photocatalysis. The absorption spectrum is dependent on the polarization direction and may be successfully regulated. The ramifications of this theoretical study for g-C_3_N_4_-based heterojunction optoelectronics and photocatalysts are important. In a nutshell, because of the band offset built-in interface polarized electric field, the energy-wasteful electron–hole recombination could be effectively reduced in the proposed g-C_3_N_4_/SnS_2_ heterostructure, and then, the photocatalytic quantum efficiencies would be improved greatly. Therefore, g-C_3_N_4_/SnS_2_ is a promising photocatalyst based on g-C_3_N_4_.

## Figures and Tables

**Figure 1 materials-18-00892-f001:**
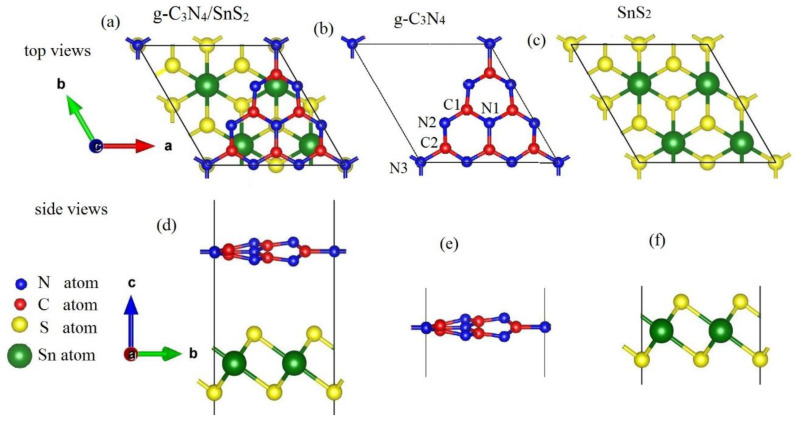
Top and side views of the optimized geometric structures of (**a**,**d**) g-C_3_N_4_/SnS_2_, (**b**,**e**) a g-C_3_N_4_ monolayer, and (**c**,**f**) SnS_2_. The blue, red, yellow, and green circles represent N, C, S, and Sn atoms, respectively.

**Figure 2 materials-18-00892-f002:**
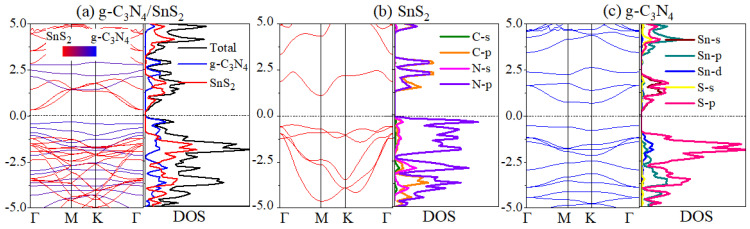
(**a**) Fat bands (**left panel**) and DOS (**right panel**) of g-C_3_N_4_/SnS_2_. (**b**,**c**) Band structures (**left panels**) and projected DOS (**right panels**) of SnS_2_ and g-C_3_N_4_, respectively. The Fermi level is at 0 eV.

**Figure 3 materials-18-00892-f003:**
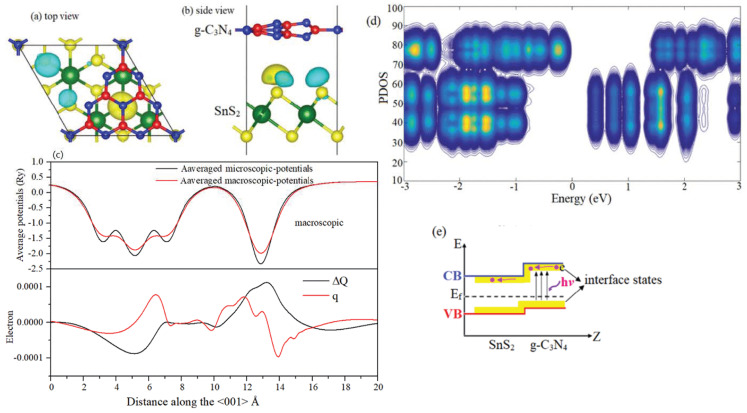
(**a**,**b**) Top and side views of charge redistribution. (**c**) Averaged microscopic and averaged macroscopic potentials (**upper panel**), the x–y plane’s average charge density difference q, and the charge displacement curve ΔQ (**lower panel**). (**d**) Local integrated PDOS along the z-direction. The PDOS values in descending order correspond to yellow, blue, and white, respectively. (**e**) Charge transfer through interface states. The blue and red bars denote the CBM and VBM for g-C_3_N_4_ and SnS_2_.

**Figure 4 materials-18-00892-f004:**
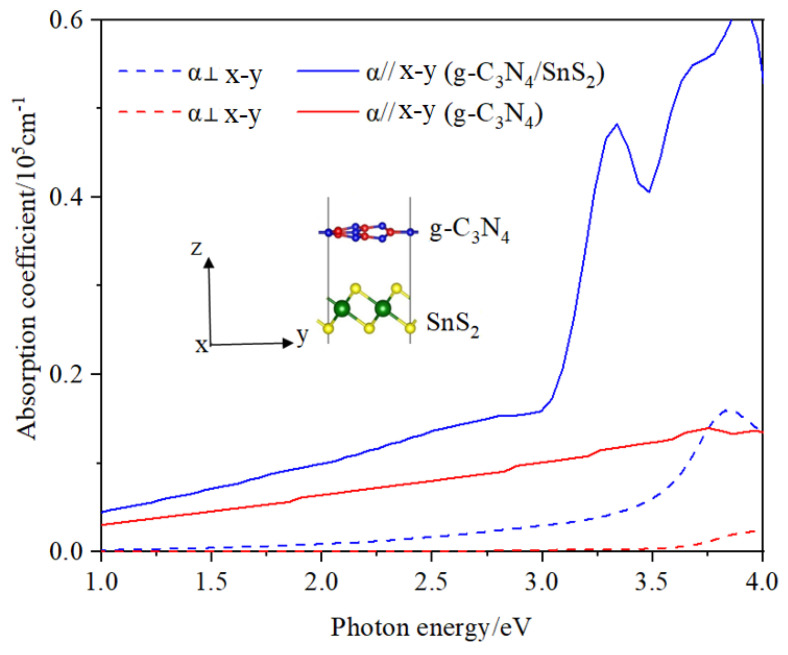
The optical absorption coefficients of g-C_3_N_4_/SnS_2_ (blue lines) and g-C_3_N_4_ (red lines) with the polarization vector α parallel and perpendicular to the x–y plane.

## Data Availability

The original contributions presented in this study are included in the article/Appendix A. Further inquiries can be directed to the corresponding author.

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
