# Peer review of "Interface Effects on the Electronic and Optical Properties of Graphitic Carbon Nitride (g-C3N4)/SnS2: First-Principles Studies"

_materials, 2025, doi:10.3390/ma18040892_

Round 1

Reviewer 1 Report

Comments and Suggestions for Authors

The work "Interface effects on the electronic and optical properties of graphitic carbon nitride (g-C3N4)/SnS2: First-principles studies" intend at studying the electronic and optical properties of the G-C3N4/SnS2 heterostructure. However, according to the literature review provided by the authors, this has been done already in several studies. On the other hand, authors keen to reveal the mechanism for enhancing the photocatalysis efficiency of the G-C3N4/SnS2 heterojunction, but this has not been done. The text contains many typos. The methods are not clear and the functional used is not suitable for the task of this work. Thus, I cannot recommend this work for publication.

The title and abstract:

The title and abstract do not correspond to the idea described in the introduction. It is said that "SnS2 and g-C3N4 band-edge potentials are found to be extremely comparable [12]" and "G-C3N4/SnS2 has been experimentally synthesized". Thus, there is no need investigate this heterostructure again. On the other hand, the authors tend to study the mechanism for enhancing the photocatalysis efficiency of the G-C3N4/SnS2 heterojunction. And this should be reflected in the title if so.

Introduction:

Some parts of the introduction are confusing. 

"Since Wang demonstrated that g C3N4 can be photocatalytically oxidized and create H2 under visible light [4]" - Ref. 4 does not belong to authors with the family name Wang.

"As a result, several researchers, including g-C3N4 functionalization and chemical doping, have concentrated their efforts..." - this sentence is confusing.

The last paragraph of the introduction is too general. According to the text, the authors study the photocatalytic efficiency of heterogeneous photocatalysts of all existing heterostructures, while in this work the g-C3N4/SnS2 heterostructure is studied.

Methods:

- When studying the band alignment, the generalized gradient approximation (GGA) functional is not suitable because it tends to significantly underestimate the band gap for semiconducting compounds. More accurate functionals such as HSE functional should be used.

- Where the calculations spin polarized? Why?

- Have the authors carried out energy cut-off convergence computations? Have they checked the Brillouin zone meshing convergence?

Results:

"The heterojunction system's lattice constant and the minimum spacing between two monolayers have been tuned to be 7.18 and 3.06, respectively" - How was this calculated? The results should be discussed in more detail and the calculation results included in supporting information.

- "The heterojunction's interfacial formation energy is predicted to be -0.027eV, demonstrating improved thermodynamic stability" - How so? Such a value of the formation energy does not mean thermodynamic stability.

- In addition, to determine the stability of the heterostructure its phonon spectrum should be calculated.

- On page 4, it is said that "little charge accumulates near the g-C3N4 layer side,". However, the CDD plot does not show this. 

- Why Average Potentials in Figure 3c are shown in Ry? VASP software, which is mentioned in the methods, output results in eV.

- Why there are two d) subfigures in Figure 3? The figure d) in the top right corner should be discussed in more detail.

- How where optical properties calculated?

- Generally, the results do not reveal the mechanism for enhancing the photocatalysis efficiency of the G-C3N4/SnS2 heterojunction.

Conclusions:

Conclusions should contain a discussion with practical application of the results.

Comments on the Quality of English Language

The English could be improved to more clearly express the research.

Author Response

Introduction:

Comment 1: Some parts of the introduction are confusing. "Since Wang demonstrated that g-C3N4 can be photocatalytically oxidized and create H2 under visible light [4]" - Ref. 4 does not belong to authors with the family name Wang.

Response 1: We are very sorry for our incorrect writing and it is rectified in the revised manuscript, where ref. 4 belong to authors with the family name Wang.

Comment 2: "As a result, several researchers, including g-C3N4 functionalization and chemical doping, have concentrated their efforts..." - this sentence is confusing.

Response 2: We appreciate your attention to the linguistic aspects of our manuscript. We understand that clear and concise communication is essential for scientific writing. In response, we have carefully revised the language paying attention to sentence structure vocabulary choices and tone. The level of English has been corrected throughout the manuscript. In addition, we have asked one colleague who is skilled authors of English language papers to check the English. We believe that these improvements will make our paper more accessible and engaging for readers.

"As a result, several researchers, including g-C3N4 functionalization and chemical doping, have concentrated their efforts..." - this sentence has been revised as below:

As a result, several researchers, including those concerned with g-C3N4 functionalization and chemical doping, have concentrated their efforts on improving electron–hole separation and visible optical absorption, which are crucial for improving catalytic efficiency.

Comment 3: The last paragraph of the introduction is too general. According to the text, the authors study the photocatalytic efficiency of heterogeneous photocatalysts of all existing heterostructures, while in this work the g-C3N4/SnS2 heterostructure is studied.

Response 3: Thank you for your suggestion. In the revised manuscript, we have re-written the last paragraph of the introduction in detail as follows:

Inspired by this experimental idea, we attempt to study the hybridization between the SnS2 sheet and g-C3N4. To design and increase the photocatalytic effectiveness of the g-C3N4/SnS2 heterostructure, the interfacial effect on the structural, electronic and optical properties must be studied. Herein, based on density functional theory (DFT) calculations, we try to investigate the interfacial interaction between monolayer g-C3N4 and SnS2 sheet and band structures of g-C3N4 and SnS2. The computation results show a type-II band alignment, which is desirable for visible light water splitting. Moreover, due to the charge transfer between g-C3N4 and SnS2 sheet, it can further enhance the photogenerated electron-hole separation. These results suggest that SnS2 sheet would be a nice choice to improve the photocatalytic performance of g-C3N4 photocatalyst.

Methods:

Comment 4: When studying the band alignment, the generalized gradient approximation (GGA) functional is not suitable because it tends to significantly underestimate the band gap for semiconducting compounds. More accurate functionals such as HSE functional should be used.

Response 4: We sincerely appreciate the valuable comments. The authors concur with the reviewer that it will be more accurate to use the HSE functional. However, due to the tight deadline of the revision, it is impractical to conduct the HSE calculations at present. Looking ahead, the authors will place more emphasis on the accurate functionals. Thank you again for your positive comments and valuable suggestions to improve the quality of our manuscript.

Comment 5: Where the calculations spin polarized? Why?

Response 5: Thank you for pointing this out. Although we agree that it is an important consideration to investigate the calculations spin polarized, this would not be possible because present calculations without spin polarized is enough to describe the electron structure of the g-C3N4/SnS2 heterostructure.

Comment 6: Have the authors carried out energy cut-off convergence computations? Have they checked the Brillouin zone meshing convergence?

Response 6: Yes, it is necessary to test the cut-off energy and K mesh grid before the relevant calculation. In our work, for g-C3N4/SnS2 heterojunction, we gradually increase the cut-off energy and K mesh grid until the energy curve becomes flat, where the cut-off energy and K mesh are 520 eV and 7×7×1, respectively. We believe that these parameters can ensure the convergence of the electronic structure of the heterojunction system.

Results:

Comment 7: "The heterojunction system's lattice constant and the minimum spacing between two monolayers have been tuned to be 7.18 and 3.06, respectively" - How was this calculated? The results should be discussed in more detail and the calculation results included in supporting information.

Response 7: Thanks for your suggestions. In the revised manuscript, we have added description and discussion in more detail for the relevant calculations as below.

In our work, we perform benchmark calculations for the isolated single-layer g-C3N4 and SnS2 sheets to examine the reliability of the computational parameters. After fully relaxation, the lattice parameters are a=b= 7.15Å for g-C3N4, and a=b=3.70 Å for SnS2, respectively. To build g-C3N4/SnS2 heterojunction, we adopt 1×1 and 2×2 supercells for g-C3N4 (6 C and 15 N atoms) and SnS2 (including 4 Sn and 8 S atoms), respectively, as shown in Figure 1. For g-C3N4 and SnS2, the mismatch is 3% for x and y directions, and a vacuum spacing of 15 is used to reduce artificial contact. For this heterogeneous structure, we perform fully structure relaxation, and the lattice parameter is a= b= 7.18 Å, meanwhile the minimum spacing between two monolayers is 3.06 Å. The bond lengths of C1-N1, C2-N2, C3-N3, SnS1, and Sn-S2 are 1.490, 1.353, 1.348, 2.568, and 2.567 Å, respectively. (page 3, paragraph 1)

Comment 8: "The heterojunction's interfacial formation energy is predicted to be -0.027eV, demonstrating improved thermodynamic stability" - How so? Such a value of the formation energy does not mean thermodynamic stability.

Response 8: The authors have explained what is the formation energy of a chemical reaction in detail below:

The term “formation energy” is commonly used in science to refer to the “formation enthalpy” (also known as the standard formation enthalpy), which is an important concept in chemical thermodynamics. Formation enthalpy represents the change in energy required to produce a chemical substance under standard conditions (usually 25°C and 1 atmosphere).

  1. Definition of formation energy

Formation enthalpy is defined as the amount of energy released or absorbed when one mole of a compound is produced from the most stable form of the monomer. The unit of formation enthalpy is usually kilojoules per mole (kJ/mol). The symbol for the formation enthalpy is usually ΔHf.

(a) Standard Formation enthalpy (ΔHf): The formation enthalpy required to produce a substance under standard conditions (25° C and 1 atmosphere).

(b) Standard formation enthalpy of a monomer is zero: The standard enthalpy of formation of an element in its most stable monomer form (e.g., N2, O2) is zero. This is because in this state no formation reactions are involved.

  1. Calculation and application of energy of formation

The formation enthalpy can help to calculate the total energy change in a chemical reaction. For example, according to Hess's Law, we can calculate the enthalpy change of a complex reaction from the formation enthalpy of each substance.

The following equation can be used to calculate the enthalpy change of a reaction (ΔH reaction):

ΔH reaction=∑ΔHf (products)-ΔHf (reactants)

The formation enthalpy is positive or negative, indicating the exothermic or absorptive nature of the reaction.

Positive value: the formation enthalpy is positive (absorbing heat), which means that energy needs to be absorbed to produce the substance.

Negative value: the formation enthalpy is negative (exothermic), which means that energy will be released when the substance is formed.

  1. The role of formation enthalpy in thermodynamics

Formation enthalpy is an important parameter for determining the stability of a substance. In thermodynamics, the enthalpy of formation is usually used together with the Gibbs Free Energy to evaluate the stability of a substance.

In thermodynamics, enthalpy of formation is often used in conjunction with Gibbs Free Energy to assess the spontaneity of a reaction.

Spontaneous reactions: the enthalpy of generation is negative and favors a decrease in Gibbs Free Energy. Exothermic

Non-spontaneous reaction: the enthalpy of generation is positive and additional energy input is required for the reaction to proceed. Absorption of heat

Comment 9: "In addition, to determine the stability of the heterostructure its phonon spectrum should be calculated.

Response 9: Thank you for your suggestion. In supporting information Figure S1, we have added the phonon spectrum to elucidate the stability of the heterostructure.

Comment 10: "On page 4, it is said that "little charge accumulates near the g-C3N4 layer side,". However, the CDD plot does not show this.

Response 10: The authors apologize for causing confusion to the reviewer. In the original manuscript, "little charge accumulates near the g-C3N4 layer side" means "no charge accumulates near the g-C3N4 layer side". We have corrected "little charge accumulates near the g-C3N4 layer side" to "no charge accumulates near the g-C3N4 layer side" in the revised manuscript. (page 4, the last line)

Comment 11: "Why Average Potentials in Figure 3c are shown in Ry? VASP software, which is mentioned in the methods, output results in eV.

Response 11: Because Average Potentials in Figure 3c are shown in Ry will show the main figure more clearly. And the OUTCAR files of VASP software output result in eV.

For example, the energy calculation result of an arbitrary OUTCAR file is as follows:

Comment 12: Why there are two d) subfigures in Figure 3? The figure d) in the top right corner should be discussed in more detail.

Response 12: We feel sorry for our carelessness. In the revised manuscript, Figure 3 has been improved: the (d) notation on the ΔQ-q diagram has been deleted as shown in the figure below. And the figure d) in the top right corner have been discussed in more detail. (page 5, paragraph 2, lines 2-14).

Comment 13: How where optical properties calculated?

Response 13: To investigate the optical properties of g-C3N4/SnS2, the optical absorption spectra are calculated by converting the complex dielectric function to the absorption coefficient α by the following equation, where and  are the real and imaginary absorption factors, respectively.

The above has been added to the revised manuscript (page 3, the last paragraph), and the specific steps are as follows:

Step 1: Static self-consistent calculation to get WAVECAR file.

(1) After completing the structural optimization, copy its CONTCAR, POTCAR files to the folder 2scf for the static calculations; and copy the script vasp.slurm for the submission calculations to 2scf.

(cp CONTCAR POTCAR ../2scf

cp vasp.slurm ../2scf)

(2) Go to the 2scf folder and copy CONTCAR as POSCAR.

(cd ../2cf

cp CONTCAR POSCAR)

(3) KPOINTS are generated using vaspkit, using the Gamma method with the spacing set to 0.02 (optical property calculations require the use of denser k-points).

(4) Use vaspkit to generate INCAR, and modify it as follows:

add SYSTEM, ISPIN=2, MAGMOM, ENCUT=520, SIGMA=0.01. in INCAR file. Note that LWAVE = .TRUE., so as to be able to generate the WAVECAR file to be used in the next step.

(5) Check that INCAR, POSCAR, POTCAR, KPOINTS, vasp.slurm are all present, then submit job.

Step 2: the frequency-dependent dielectric function and optical property calculations based on the Independent-Partical Approximation (IPA)

(1) The reference is the calculation of the frequency dependent dielectric function from the manual on the vasp website - check the OUTCAR file in the 2scf folder for the value of NBANDS, and subsequent NBANDS are set to 2 to 3 times this value. (grep NBANDS OUTCAR)

(2) Create a new ipa-optical folder and copy WAVECAR KPOINTS POTCAR POSCAR vasp.slurm from the 2scf folder to the ipa-optical folder.

(mkdir ../ipa-optical

cp WAVECAR KPOINTS POTCAR POSCAR vasp.slurm ../ipa-optical)

(3) Use vaspkit to generate the INCAR file by typing vaspkit-1-101-OP at the command line in order to get INCAR and INCAR.step2.lef.

(4) The first of these is from the vasp official manual for frequency-dependent dielectric function calculations on the independent-particle picture subsection, which is based on the independent-particle assumption. Second, we should take into account local field effects, is from the including local field effects subsection. The second calculation can only be performed if the first calculation is based on the independent-particle assumption (and LPEAD=.TRUE. in INCAR). This subsection performs the first calculation, referred to as IPA.

(5) INCAR is modified as follows, ISPIN MAGMOM ENCUT is added, NBANDS=96, CSHIFT=0.15 (the larger the value, the smoother the curve), LPEAD=.TRUE.(WAVEDER file is generated).

Check that INCAR, WAVECAR, POSCAR, POTCAR, KPOINTS, vasp.slurm are all present, and submit the assignment.

(6) Plot dielectric function by gnuplot using official website scripts.

(7) Use the vaspkit 711 command to get the optical coefficients such as EXTINCTION.dat and ABSORPTION.dat.

(8) Draw a plot using origin.

Comment 14: Generally, the results do not reveal the mechanism for enhancing the photocatalysis efficiency of the G-C3N4/SnS2 heterojunction.

Response 14: Thank you for your suggestion. In the revised manuscript, we have re-written section 3.3 to reveal the mechanism for enhancing the photocatalysis efficiency of the G-C3N4/SnS2 heterojunction. (page 5, paragraph 2).

Conclusions:

Comment 15: Conclusions should contain a discussion with practical application of the results.

Response 15: Thank you for your suggestion. In the revised manuscript, we have re-written conclusions containing a discussion with practical application of the results. (page 6, the last paragraph).

Reviewer 2 Report

Comments and Suggestions for Authors

The manuscript can be accepted after minor revision according to following suggestions:

1. Introduction 

- 3rd sentence: : „Since Wang demonstrated…” The refered [4] does not include Wang.

2. Computational method

- 3rd sentence: eV is doubled

- Between equation (1) and (2): 4th sentence „ The he integrated…” The „he” is unnecesarry.

3.2 Fat bad and density of states (DOS)

- 4th sentence: The Authors refer to Fig 2 (c) and (d). But there is no (d) part in Fig. 2.

3.3. Charge transport

- 1st sentence: delete „flowing” from the expression „following flowing reasons…”

- Fig 3.: The (d) notation is unnecessary on the deltaQ-q diagram.

3.4. Optical properties

It is not clear for me that the spectra shown in Fig. 4. belong to real (prepared) sample or they are originated from calculation. If the spectra were measured, the preparation method should be mentioned.

Author Response

  1. Introduction

Comment 1: 3rd sentence: : „Since Wang demonstrated…” The refered [4] does not include Wang.

Response 1: We are very sorry for our incorrect writing and it is rectified in the revised manuscript, where ref. 4 belong to authors with the family name Wang.

  1. Computational method

Comment 2: 3rd sentence: eV is doubled

Response 2: Thank you for pointing out the error in our manuscript. We have deleted the superfluous eV in the revised manuscript.

Comment 3: Between equation (1) and (2): 4th sentence „ The he integrated…” The „he” is unnecesarry.

Response 3: Thank you for your suggestion. In the revised manuscript, The „he” between equation (1) and (2): 4th sentence „ The he integrated…” has been deleted.

3.2 Fat bad and density of states (DOS)

Comment 4: 4th sentence: The Authors refer to Fig 2 (c) and (d). But there is no (d) part in Fig. 2.

Response 4: We are very sorry for our incorrect writing. In the revised manuscript, the present Figs. 2 (c) and (d) have been replaced by Figs. 2 (b) and (c). (page 4, paragraph 1, line 10)

3.3. Charge transport

Comment 5: 1st sentence: delete „flowing” from the expression „following flowing reasons…”

Response 5: Thank you for your suggestion. In the revised manuscript, “following flowing reasons” has been deleted. (page 4, the last paragraph 1, line 1)

Comment 6: Fig 3.: The (d) notation is unnecessary on the delta Q-q diagram.

Response 6: Thank you for your suggestion. In the revised manuscript, Fig. 3 has been improved: the (d) notation on the ΔQ-q diagram has been deleted as below.

3.4. Optical properties

Comment 7: It is not clear for me that the spectra shown in Fig. 4. belong to real (prepared) sample or they are originated from calculation. If the spectra were measured, the preparation method should be mentioned.

Response 7: The authors apologize for causing confusion to the reviewer. In the revised manuscript, the spectra shown in Fig. 4 is originated from calculation which has been mentioned in the first in the first sentence of section 3.4. (The photocatalytic efficiency of the g-C3N4 monolayer and g-C3N4/SnS2 has been calculated using the absorption spectra shown in Fig. 4.)

Reviewer 3 Report

Comments and Suggestions for Authors

The work was reviewed, and the following suggestions are made to expand the information:

 Place the information of the g-C3N4/SnS2 composite, such as the load of g-C3N4 placed on the SnS2, coupling method for these semiconductors, whether the semiconductors or the salts of the precursors were used for the synthesis of both semiconductors, etc.

 If you have the information on the electrochemical characterization of the g-C3N4/SnS2 composite, use it to first look at the modifications that the SnS2 underwent when doped with g-C3N4 and know the VB and CB values.

The composite was excited by an irradiation source to obtain some of the results reported in the work, detailing the lighting source used and what would happen if it were to be changed for another type of lamp, for example, if a UV lamp was used what happens if it is changed for one that only emits in the visible region or vice versa.

In Fig. 4, it is mentioned that g-C3N4 has a signal at 3.3 eV. However, in Fig. 4, the signal at 3.3 eV is for the g-C3N4/SnS2 composite. Which one is correct?

 The work mentions that they are first-principles studies, and these were made into a g-C3N4/SnS2 composite whose physical, morphological, electrical, and optical characteristics come from the synthesis method used. It was studied if a value such as the load of g-C3N4 in the SnS2 is changed. What behavior did change in the characterizations made in this work follow? Whether they remained the same, decreased, or increased.

Author Response

Comment 1: Place the information of the g-C3N4/SnS2 composite, such as the load of g-C3N4 placed on the SnS2, coupling method for these semiconductors, whether the semiconductors or the salts of the precursors were used for the synthesis of both semiconductors, etc.

Response 1: Thank you for the suggestion. The authors totally agree with the reviewer. In the revised manuscript, we have placed the information of the g-C3N4/SnS2 composite which are described below:

Inspired by this experimental idea, we attempt to study the hybridization between the SnS2 sheet and interfacial effect on the structural, electronic and optical properties must be studied. Herein, based on density functional theory (DFT) calculations, we try to investigate the interfacial interaction between monolayer g-C3N4 and SnS2 sheet and band structures of g-C3N4 and SnS2. The computation results show a type-II band alignment, which is desirable for visible light water splitting. Moreover, due to the charge transfer between g-C3N4 and SnS2 sheet, it can further enhance the photogenerated electron-hole separation. These results suggest that SnS2 sheet would be a nice choice to improve the photocatalytic performance of g-C3N4 photocatalyst.

Comment 2: If you have the information on the electrochemical characterization of the g-C3N4/SnS2 composite, use it to first look at the modifications that the SnS2 underwent when doped with g-C3N4 and know the VB and CB values.

Response 2: Thank you for the suggestion. In the revised manuscript, we have added description and discussion in more detail for the information on the electrochemical characterization of the g-C3N4/SnS2 composite, use it to first look at the modifications that the SnS2 underwent when doped with g-C3N4 and know the VB and CB values. (page 4, paragraph 1).

Comment 3: The composite was excited by an irradiation source to obtain some of the results reported in the work, detailing the lighting source used and what would happen if it were to be changed for another type of lamp, for example, if a UV lamp was used what happens if it is changed for one that only emits in the visible region or vice versa.

Response 3: I am grateful to you for your valuable suggestion. However, because of the limitation of our present laboratory, experimental verification is hard to carry out. We are trying to improve our experimental condition, the comprehensive experiments would be obtained and reported in future.

Comment 4: In Fig. 4, it is mentioned that g-C3N4 has a signal at 3.3 eV. However, in Fig. 4, the signal at 3.3 eV is for the g-C3N4/SnS2 composite. Which one is correct?

Response 4: The authors apologize for causing confusion to the reviewer. In the original manuscript, the peak energy of g-C3N4 is around 3.3 eV, as shown in the image. (page 6, paragraph 1, lines 2-3)

Comment 5: The work mentions that they are first-principles studies, and these were made into a g-C3N4/SnS2 composite whose physical, morphological, electrical, and optical characteristics come from the synthesis method used. It was studied if a value such as the load of g-C3N4 in the SnS2 is changed. What behavior did change in the characterizations made in this work follow? Whether they remained the same, decreased, or increased.

Response 5: Thank you for your professional review work on our article. As you are concerned, any change to the g-C3N4 or SnS2 will cause some new features. However, due to the tight deadline of the revision, it is impractical to conduct the calculations by changing different values at present, and judge whether they remained the same, decreased, or increased. Looking ahead, the authors will place more emphasis on studying the change in the characterizations.

Reviewer 4 Report

Comments and Suggestions for Authors

This manuscript deals with a fundamental analysis related with the electronic behavior of a semiconductor-heterojunction based on (g-C3N4)/SnS2. As such, only simulation results are presented in this manuscript. Although the subject is interesting, some observations are made here before a possible publication in Materials journal.

a) In the text, is indicated that from simulation a thermodynamic stability is approached…How could the authors demonstrate such thermodynamic stability of the material in study from an experimental point of view??? In this context, what could be the effect of the solution pH during illumination and dark conditions-experiments (???). Did the authors consider the effect of solution concentration and amount of (g-C3N4) versus the amount of SnS2 (???) is there an effect??? Polarization of the material in a substrate such an ITO could be possible??? This array could give more information about the surface state during polarization or even during FRA… please, detailed discussions should be included in the text…

b) What kind of redox process could be carried out at the semiconductor interface??? Reduction of CO2, production of hydrogen, organics oxidation (???)… please deeper details should be included in the text… with some reactions as an example…

c) Please a detailed explanation (from a physical-experimental point of view) concerning the potential (microscopic and macroscopic) mentioned in section 3.3 should be explained in a deeper form… what is the effect of such potential during illumination in a given wavelength???

d) Please check the labels in Figure 3… label (e) is missing…

Author Response

Comment 1: In the text, is indicated that from simulation a thermodynamic stability is approached…How could the authors demonstrate such thermodynamic stability of the material in study from an experimental point of view??? In this context, what could be the effect of the solution pH during illumination and dark conditions-experiments (???). Did the authors consider the effect of solution concentration and amount of (g-C3N4) versus the amount of SnS2 (???) is there an effect??? Polarization of the material in a substrate such an ITO could be possible??? This array could give more information about the surface state during polarization or even during FRA… please, detailed discussions should be included in the text…

Response 1: I am grateful to you for your valuable suggestion. The authors concur with reviewer that additional studies, data, or further experimental results would be beneficial. However, because of the limitation of our present laboratory, experimental verification is hard to carry out. We are trying to improve our experimental condition, the comprehensive experiments would be obtained and reported in future.

Comment 2: What kind of redox process could be carried out at the semiconductor interface??? Reduction of CO2, production of hydrogen, organics oxidation (???)… please deeper details should be included in the text… with some reactions as an example…

Response 2: Thank you for the suggestion. In the revised manuscript, we have added description and discussion in more detail for at the semiconductor interface as below. (page 5, paragraph 2, lines 2-14).

When a photon of light is absorbed by g-C3N4, it energizes electrons and holes, which are then negatively transported down the CBM and VBM. That is to say, the spontaneous interfacial charge transfer from the g-C3N4 to SnS2 can be imply rationalized in terms of the large difference between two layers, which results in a built-in electric field at the interface, promoting photogenerated electrons from the CB of SnS2 to the CB of g-C3N4, and the improving the transformation photogenerated holes of the VB from g-C3N4 to SnS2. Therefore, the photogenerated electron-hole separation can be enhanced effectively by this built-in interface electric field.

Comment 3: Please a detailed explanation (from a physical-experimental point of view) concerning the potential (microscopic and macroscopic) mentioned in section 3.3 should be explained in a deeper form… what is the effect of such potential during illumination in a given wavelength???

Response 3: Thank you for the suggestion. In the revised manuscript, we have added explanation in detail for the potential (microscopic and macroscopic) mentioned in section 3.3. (page 4, the last paragraph, lines 7-11; page 5, the first paragraph, lines 1-6.)

Comment 4: Please check the labels in Figure 3… label (e) is missing…

Response 4: The authors are sorry for our careless mistakes. In the revised manuscript, label (e) has been added in Figure 3.

Round 2

Reviewer 1 Report

Comments and Suggestions for Authors

The authors have addressed most of the comments by the reviewer and therefore, this paper may be suitable for publication.